# "I don't see a reason why we should be hidden from view": Views of a convenience sample of people living with HIV on sharing HIV status data in routinely collected health and care databases in England

Elizabeth Ford[1]*, Katie Goddard[1], Michael Smith[2], Jaime Vera[2,3]

1 Department of Primary Care and Public Health, Brighton and Sussex Medical School, Brighton, United Kingdom, 2 University Hospitals Sussex NHS Foundation Trust, Brighton, United Kingdom, 3 Department of Global Health and Infection, Brighton and Sussex Medical School, Brighton, United Kingdom

* E.m.ford@bsms.ac.uk

**Editor:** Sebastian Suarez Fuller, University of Oxford Nuffield Department of Clinical Medicine: University of Oxford Nuffield Department of Medicine, UNITED KINGDOM OF GREAT BRITAIN AND NORTHERN IRELAND

## Abstract

### Introduction

People living with HIV (PLWH) now have near-normal life-expectancy, but still experience stigma, and HIV status is treated as sensitive health information. When UK healthcare patient data is curated into anonymised datasets for research, HIV diagnostic codes are stripped out. As PLWH age, we must research how HIV affects conditions of ageing, but cannot do so in current NHS research datasets. We aimed to elicit views on HIV status being shared in NHS datasets, and identify appropriate safeguards.

### Methods

We conducted three focus groups with a convenience sample of PLWH recruited through HIV charities, presenting information on data governance, data-sharing, patient privacy, law, and research areas envisaged for HIV and ageing. Each focus group involved two presentations, a question session, and facilitated breakout discussion groups. Discussions were audio-recorded, transcribed and analysed thematically.

### Results

37 PLWH (age range 23-58y) took part. The overarching theme was around trust, both the loss of trust experienced by participants due to previous negative or discriminatory experiences, and the need to slowly build trust in data-sharing initiatives. Further themes showed that participants were supportive of data being used for research and health care improvements, but needed a guarantee that their privacy would be protected. A loss of trust in systems and organisations using the data, suspicion of data users' agendas, and worry about increased discrimination and stigmatisation made them cautious about data sharing. To rebuild trust participants wanted to see transparent security protocols, accountability for

**Data Availability Statement:** The data underlying this study have been deposited in the University of Sussex Figshare Data Repository and can be accessed at https://doi.org/10.25377/sussex.26969746.v1. Fully anonymized transcripts are freely available. More sensitive transcripts are accessible upon reasonable request to Brighton and Sussex Medical School by contacting the research team via email at primarycareda@bsms.ac.uk.

**Funding:** This work was funded by the University of Sussex Healthy Ageing Participatory Research Fund (awarded to EF, JV and MS) and the National Institute of Health and Care Research (NIHR) Applied Research Collaboration in Kent, Surrey and Sussex (NIHR200179) (supporting the salary of EF and KG). The funders had no role in the design of the study and collection, analysis, and interpretation of data and in writing the manuscript.

**Competing interests:** The author(s) declared no potential conflicts of interest with respect to the research, authorship, and/or publication of this article.

following these, and communication about data flows and uses, as well as awareness training about HIV, and clear involvement of PLWH as full stakeholders on project teams and decision-making panels.

## Conclusions

PLWH were cautiously in favour of their data being shared for research into HIV, where this could be undertaken with high levels of security, and the close involvement of PLWH to set research agendas and avoid increased stigma.

## Introduction

The successful introduction of combination antiretroviral therapy (cART) has transformed human immunodeficiency virus (HIV) disease into a chronic condition where most people living with HIV (PLWH) on effective treatment have a near normal life expectancy [1]. Despite this success, perceived HIV stigma and discrimination remains a significant barrier for PLWH accessing services and novel health initiatives that could improve their health outcomes [2–4]. To protect PLWH from stigma there has been a reluctance to include these patients in data sharing or data mining initiatives due to the fear of disclosure of HIV status.

Due to the stigma around HIV and other sexually transmitted infections, there used to be strict regulations around the sharing of this data in the United Kingdom (UK), historically governed by the National Health Service (NHS) (Venereal Diseases) Regulations 1974 and the NHS Trusts and Primary Care Trusts (Sexually Transmitted Diseases) Directions 2000. The regulations stated that information concerning sexual health should not be shared without explicit patient consent, even when this may affect patient care. In the 2012 (and updated 2022) Health and Social Care Act, HIV status and other sexual health data was not treated as special category data any longer, but all health data was seen as equally sensitive. However, due to the legacy of stigma around HIV and other STIs, sexual health information is often stored separately from other health data, and general practitioners (GPs) and hospital records do not have access to sexual health clinic data. Due to the implications for their wider health, PLWH in the UK are strongly encouraged to inform their GP of their HIV status, which then may be recorded in their GP patient record. UK studies estimated that 50.7% of HIV patients had their HIV status recorded by their GP in 2005, 73% of PWLH had their diagnosis in GP data in 2010, and in 2015, 86% of survey respondents had shared their data with their GP, suggesting an upward trend in disclosure [5–7].

Information collected throughout routine patient health care in the UK NHS is increasingly extracted into de-identified databases for audit and research purposes, used both within healthcare organisations (HCOs) and made available to external users such as university researchers; this is increasingly true for sexual health research [8–11]. While patient identifiers are removed, datasets include demographic data, the patient's health conditions and diagnoses, symptoms, tests, referrals, procedures, stays in hospital, prescription and administrative (e.g. number and type of consultations) data [12]. Due to the large volume of patient data held in these databases, they are very effective for monitoring health trends among a population of patients with relatively rare conditions. In comparison, classical data collection methods are less able to capture such information, as patients are spread across a large geographical area [13–15]. Because everyone in the UK has a right to register with a GP practice [16], and 98% of the population are registered [17], NHS datasets also include data on populations who may not ordinarily come forward for research participation, and therefore can inform research on

the healthcare outcomes of seldom heard groups. In addition, especially in primary care or general practice data, data is collected on patients over many years, and can therefore be used to examine longitudinal trends in health conditions and medication usage, as in a 2022 study looking at the use of Pre-exposure Prophylaxis (PrEP) for HIV [18].

Due to legacy concerns regarding the sensitivity of HIV data, patients' HIV status is mostly not uploaded to datasets of patient records during the data extraction process, either because sexual health clinics do not contribute data to these linked databases or because HIV diagnosis codes in GP data are redacted at the point of data extraction. It is therefore not currently possible to use these datasets to understand health trends in PLWH across their lifespan. This is particularly concerning as PLWH are increasingly affected by chronic conditions typically associated with ageing such as cardiovascular disease, bone disease, and cognitive impairment [19, 20]. The suppression of HIV data in datasets of routinely collected NHS health data represents a barrier to understanding the health problems that PLWH are facing, and this leads to health inequity for PLWH.

Overturning current governance procedures unilaterally, to reduce this inequity, would risk losing trust of PLWH in their medical confidentiality and in wider research endeavours. Instead, by using a participatory approach, we aimed to encourage PLWH to set the agenda on whether their HIV status should be shared for research, and what the safeguards around this data-sharing should look like.

After deliberative engagement on health data-sharing, most UK public participants are favourable to sharing their data as long as data is anonymised, kept secure, and used only for public benefit [21, 22]. However, it is not clear whether these principles are also held by PLWH. To start a participatory process to inform data governance decisions around HIV status data, we organised deliberative online focus groups with PLWH to address the following questions:

1. What are the views of PLWH on their health data being included in databases of routinely collected health data?

2. What safeguards do PLWH want to see surrounding their data, if it is to be stored in databases of routinely collected health data?

3. For what purposes would PLWH want their data to be used?

## Methods

### Inclusion criteria

We included any adult living in the UK, who had a self-reported diagnosis of HIV, spoke and understood English well enough to fully participate in an online discussion via Zoom, and was able to attend one of the available days for the focus groups.

### Recruitment

We recruited a convenience sample of participants via emails and newsletters through UK charities between 5/5/22 and 5/7/22: The National AIDS Trust, Terrence Higgins Trust and the AIDSmap. These groups both had significant reach to PLWH through websites, social media and mailing lists. We provided a flyer and a short text explanation of the project which was circulated among the members of these groups. To request further information or express interest an email address for a study team member was provided.

Following expression of interest, participants were sent an information sheet and consent form, followed by a demographics questionnaire and some pre-group written information

about data-sharing and HIV. Informed written consent was required before a participant could join the Zoom call, however, participants were told that if they wanted to remain anonymous in the group they could join the Zoom call with a pseudonym and keep their camera off. Following attendance to the group and return of the demographics questionnaire, each participant was offered a voucher for £62.50 representing a Thank You gift for a commitment of 30 min preparation time and 2 hours' focus group attendance.

## Discussion group format

We anticipated that PLWH would know little about the curation and use of routinely collected health and administrative data in the UK for service planning, evaluation and research, so we chose a research design in which we could inform and support participants to understand the different issues before deliberating on their opinions and views. This is known as deliberative research, an approach for gathering wider views on issues when there are many complex issues to weigh against each other [23, 24]. Our focus groups included two informative presentations, question and answer sessions, facilitated discussions in smaller breakout rooms and a plenary feedback session (details given in Table 1, and sample screenshots of slides in S1 File).

The information presented was balanced and comprehensive, incorporating the possible benefits and risks of sharing data on HIV status. The breakout room discussions (N = 4 to 6 participants) aimed to be inclusive of all group members while respecting participants' desire to protect their identities, thus we used both audio and written chat functions. Each focus group lasted two hours.

## Addressing digital exclusion

We deliberately organised our discussion groups online via Zoom software. This was because we wanted to offer the participants the choice of remaining anonymous during the group. Participants could join using a pseudonym and with their camera off, so that we only heard their voice. Some participants additionally contributed using the chat function.

**Table 1. Focus group structure.**

| |
|---|
| **Presentation 1—What are the current rules on HIV data sharing and what are the risks of data sharing, what mechanisms already protect data, and what additional safeguards could be put in place? (Delivered by EF)** |
| This session was led by an experienced health data researcher and looked at laws around health data sharing, described the way health data is usually kept safe, the risk of re-identification from de-identified routinely collected data and the possibilities of data breaches. It described some risks that have previously been feared by public participants around data-sharing. It then discussed the safeguards that can be placed on data which is shared to allow some activities to take place but minimise the risk of data breaches, such as data safe-havens. |
| **Presentation 2—What are the benefits of wider sharing of HIV health data? (delivered by JV or MS)** |
| This session was led by an experienced HIV researcher and discussed the current issues in caring for older people living with HIV and the opportunities for health research that could be available specifically in HIV health issues, if data was shared more openly. |
| **Breakout Discussion Questions:** |
| In breakout rooms, the following questions were asked by a facilitator: |
| 1) What benefits can you see for yourself or for other people living with HIV, of widening access to the de-identified health data of people living with HIV? |
| 2) What are the most important projects that you believe could be undertaken with widened access to HIV health data? |
| 3) What are your main concerns about sharing de-identified data on HIV status? |
| 4) What combination of safeguards, checks and oversight would mitigate your concerns about data-sharing? |
| 5) Are there any particular groups you would or wouldn't want to have access to your data? |

We recognised the risk that hosting discussion groups online may have excluded some potential participants. To reduce unintentional exclusion, we:

1. Ran our focus groups at different times of day (in working hours, and in the evening).

2. Offered that if an individual was not proficient in English, they could attend with another member of household to help with English. Due to funding constraints, we did not offer any other translation services.

3. Provided options for expressing interest and/or opinions via email, telephone and postal service.

4. To ensure accessibility to the Zoom environment, we checked each participant was familiar with how to use Zoom before the focus group, sent a Zoom user guide, and offered for the participant to join the Zoom call via the telephone.

## Data analysis

Demographic data was summarised. All focus groups were recorded using Zoom recording features which captured both audio and video recordings, and automatically transcribed audio in the "main room". Breakout rooms were recorded on an external audio device, uploaded to a secure server immediately then deleted from the device. Audio files were sent to an external transcribing company for verbatim transcription with removal of all identifiers. Zoom automated transcripts were checked and corrected for quality and anonymity against the original audio files by KG before analysis. "Chat" files were either downloaded or copied from the screen and stored securely.

The resulting qualitative data was analysed according to 6-step thematic analysis principles outlined by Braun and Clark [25–27] using NVivo software between 1/12/22 and 1/2/23; authors had no access to identifiable participant information at this time. First, one researcher (EF) familiarised herself with the data by reading and re-reading transcripts and noting down initial ideas. The same researcher coded the data (EF), highlighted relevant features of the data and assigned them codes, then applied codes in a systematic fashion across the entire dataset, this process created simple "bucket" topic codes. A second researcher (KG) checked the data, coded additional data within the coding structure, and had the opportunity to suggest further codes, but did not identify any more. Codes were then collected into potential themes, and themes were then discussed among all members of the research team and mapped and adjusted. The raw data was again revisited and in conjunction with the coded data, and while writing about each potential theme, the analyst looked for patterns of shared meaning, or latent concepts, underpinning the different topics. Following reflection on the commonality of underlying messages, exploring ways to interpret these, and further thoughtful engagement, going back and forth with the data, a hierarchy of themes was reached. This process was both inductive and deductive, because the researcher was both somewhat constrained by the topics explored in the discussion groups, but also seeking for novel themes within the data without expectation. The researcher was aware of being a white female academic without lived experience of HIV and therefore reflected on the need to centre the voice of the participants and was cautious, due to lack of lived experience, not to over-ascribe intention or meaning beyond what was found in the data. After careful reflection and checking within the team, the resultant analysis offered some interpretation and weaving together of the ramifications of seminal experiences of the participants, while trying to avoid "speaking over them". For this reason, multiple extracts and quotes were selected to illustrate each theme; these quotes are presented to centre the participant voice in this research.

### Ethical approvals

Ethical approval was given by the Brighton and Sussex Medical School Research Governance and Ethics Committee (ER/JV95/14).

## Results

### Participant characteristics

Thirty-seven people living with HIV took part in the focus groups, of whom 36 supplied demographic data, shown in Table 2. The median age was 26 years with a range of 23-58y. Twenty-seven were male, 8 were female and one gave their gender as transgender. All were living in the UK; 35 were in England and one in Scotland. Participants lived in predominantly urban areas, with 13 based in London and a further 15 based in another UK city. Twenty-six participants were Black African or Caribbean, or mixed White and Black background, nine participants were white British, and one was from a mixed white and Asian background. It was noted by facilitators that for many participants, English was not their first language. Twenty-five participants gave their sexual orientation as straight, five were gay or lesbian, and six were bisexual.

### Themes from the discussions

One latent, or overarching, theme ran through all the subthemes found in the analysis, which was centred on the legacy among participants of loss of trust in healthcare and related organisations and systems, and the need to build this trust again for widespread support of data sharing. Nested under the overarching theme of trust, mapping to the focus of the discussions, there were three larger themes; these were named "data sharing could improve our lives", "fear of discrimination" and "accountability and awareness". Each of these had several subthemes, as shown in Fig 1.

The overarching theme within the data was trust. People living with HIV had had negative experiences which had led to them losing trust in organisations' motivations to hold their data,

**Table 2. Participant demographic characteristic.**

| Characteristic | | N = 36 (1 missing) |
|---|---|---|
| Age | Median (range), years | 26 (23–58) |
| Sex/gender | Male | 27 (75%) |
| | Female | 8 (22%) |
| | Transgender | 1 (3%) |
| Ethnicity | White British | 9 (25%) |
| | Black–any | 15 (42%) |
| | Mixed background: White and Black African/Caribbean | 11 (31%) |
| | Mixed background: White and Asian | 1 (3%) |
| Sexual orientation | Heterosexual | 25 (69%) |
| | Gay or Lesbian | 5 (14%) |
| | Bisexual | 6 (17%) |
| Location | England | 35 (97%) |
| | Scotland | 1 (3%) |
| | London | 13 (36%) |
| | Other UK city | 15 (43%) |
| | Smaller town/rural | 8 (22%) |

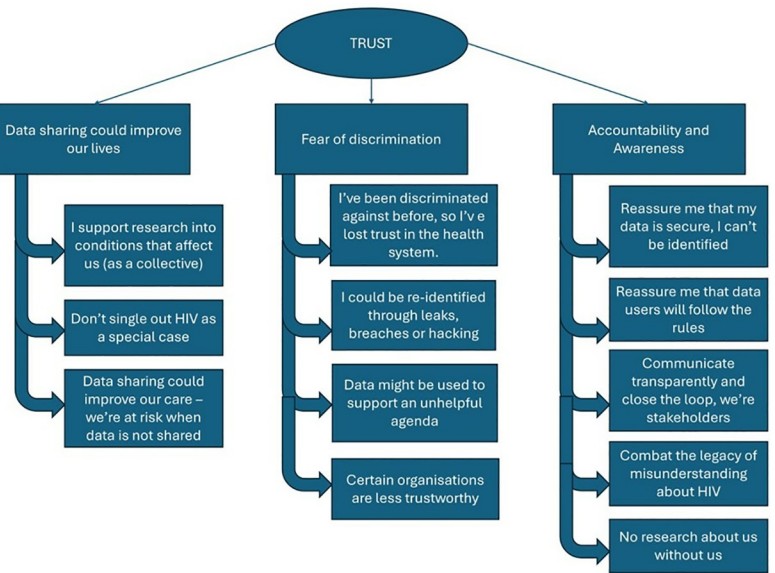

**Fig 1. Theme structure within the data.**

the intent of individuals in the organisations to help them, and the competency of organisations keeping their data secure. From a deep consideration of the data, the question emerged "how could this proposed data sharing be made trustworthy when PLWH have had so many experiences of their trust being broken?" This issue of trust was embedded in most of the subthemes, including discussions around ways of helping to rebuild the trust of PHLW in research endeavours and healthcare improvements.

## Theme 1: Data sharing could improve our lives

**1.1 I support research into conditions that affect us.**   Many of the participants expressed support for the idea of their HIV status being shared alongside their other health data, for the purposes of research or improving care. There was a collective sense that this sharing could help other patients living with HIV, and that as a community, all would be better off. The sharing of one's data could be seen as an act of altruism; a passive way of taking part in research to bring about improvements in care.

"Most of us anyway, are in agreement that . . . that the benefits to sharing all the data are . . . are huge"

(FG1 BOR1)

"I don't see any point where I, that I personally, help, so I think it's very much helpful if everyone gets to know more information about the current status."

(FG1 BOR2)

Participants hoped that by sharing their data, a range of research projects on HIV could be carried out and discussed a variety of potential research topics. These included better support for mental health, quality of life improvements, diseases of ageing, and integration of care for HIV with care for other conditions.

For mental health, participants hoped that research would suggest ways of supporting PLWH who had depression and anxiety, and to identify where PLWH were seeking mental health support.

"But you know sometimes you feel depressed and you feel sad about it, about you know being . . . positive. So, I think if there are any projects that could be done with the health data, . . . I mean social engagement to help them with the depression and anxiety, you know, just making sure there's support".

(FG2 BOR1)

"It would be really good to have some more research and some better linking up of where . . . people living with HIV are seeking psychological and mental health support though their GPs and through their IAPT [primary care psychotherapy] service."

(FG2 BOR2)

Others suggest that research on how to increase "wellbeing" or quality of life would be welcome, including reducing levels of stigma and increasing awareness, so that PLWH felt better able to cope in society.

"I think a lot of research should be done about the wellbeing of people like us because that is really the most important thing, . . .like because people of different race and colour shouldn't be restricted, shouldn't . . . that equality should be there, so research should be done concerning that."

(FG2 BOR1)

"There is something really important about how our quality of life is measured and understood—being undetectable (viral suppression) is not enough in itself; we need to be supported, including in primary care, to live well with HIV"

(FG2 Zoom chat)

"I believe that there are a lot of programmes, and a lot of research, research that are to be carried out because . . . considering the level of stigmatisation in the society with people like us that are HIV positive. So, I think that perhaps there should be research on the level of awareness . . . the society, how to cope, how to help HIV patients to cope with the stigmatisation level."

(FG2 BOR1)

We can see in these quotes the use of phrases such as "people like us" and "our quality of life", suggesting participants had a sense of belonging to a group or collective, but that this group was set apart or different from others and would have different needs in terms of research and healthcare. Participants also referred to the stigmatisation of "people like us", suggesting this stigmatisation is something they carry around with them and which determined their sense of being different from others.

Participants agreed that it was important to carry out research on the ageing process, and the increased risk of comorbidities.

"It's . . . very, very, very so important to carry out research on, you know, the ageing and the process of . . . you know, I would say the sort of life that somebody living with HIV has."

(FG2 BOR1)

"As a group of people, or group of patients, we're . . . we're getting increasingly older, and you talked about lots of other comorbidities that we . . . we might face."

(FG2 BOR2)

With comorbidities in mind, participants expressed that integrating care for multiple conditions would be a priority in terms of improving healthcare for PLWH. Participants were particularly clear that healthcare for different conditions was not joined up and often did not take account of their HIV status, because this data was not shared around the health system. Specialists in non-HIV conditions did not have good knowledge about how HIV may affect the additional condition, meaning that care was fragmented and often the patient felt it was incumbent on them to make sure their care was safe, checking with pharmacists about medication interactions, for example. Participants therefore were hoping that better data sharing might increase research on the effect of HIV, and its treatment, on other long-term conditions.

"If we have another long term condition alongside our HIV that we're living with, we can often end up playing ping pong between different specialties. So, and often it can be really, really frustrating for us as . . . as . . . as patients to not have our problems dealt with in an integrated way."

(FG2 BOR2)

**1.2 Don't single out HIV as a special case.**   Some participants, particularly those who had lived with HIV for a long time, felt that it was time to stop treating HIV as a "special" or especially private condition. They felt that they did not need to feel ashamed or guilty about the condition and that patients and all involved in healthcare provision should be open about it, as they were with any other medical condition.

"Just by kind of . . . perpetuating this singling out HIV as a medical condition, we are in effect just kind of perpetuating the stigma around it. . . it's a medical condition, you know, it's a medical condition that we've all got and . . . and the more we kind of tread around it with eggshells, the more I feel as if we're just giving fuel to the fire of the fact that we . . . should be ashamed of it, and we should feel guilty about the fact that we have HIV,"

(FG1 BOR1)

However, other participants held the view that no-one in their lives could find out about their HIV positive status, and this was particularly important medical information to keep private. They spoke about how certain types of information–headache, or flu–could be shared openly, but that they would not want HIV shared openly, whether in a clinical setting, or for research, or socially. This suggested that other participants in our focus group still saw HIV status as more private or sensitive than other types of health data.

"If someone is living with HIV, if data get shared, there should be an extra protection put on those ones. . . like I said, a flu or a pain, or a headache or something can just be open"

(FG1 BOR1).

**1.3 Data sharing could improve our care, we're at risk when data is not shared.**   Participants hoped that the sharing of data could "*improve the chances of getting better treatment in*

*hospitals*, *clinics*" (FG1 BOR1), and that sharing data would support the development of new medications, and broaden different topics for health research on HIV. Embedded within these views were the idea that the more research that was enabled, the more awareness of HIV could be fostered, so that data sharing could have a wider societal impact.

> "Increasing data sharing means that there'll be other sort of research that can be done that will help people living with HIV and . . . and, you know, in terms of medications, might make them more accessible, but also might increase awareness of HIV among governments."
>
> (FG2 BOR2)

Other participants hoped that projects on the welfare of HIV patients in society could be carried out and that there could be a change to the level of stigma experienced, as well as raising awareness of HIV to improve services. There was the idea that challenging the gaps in knowledge and building awareness within the structure of society would help move policy forward, as well as improving how citizens are treated and how HIV as a condition is 'seen'.

> "I think sharing the data will also make a lot of interest, a whole lot of people who care. 'Oh these people are living with [HIV], how are they cared for, what are the strategies, what are the things that are being done to them?' Surely, they'll tell you, they'll bring attention and care to HIV patients, and people living with HIV."
>
> (FG1 BOR2)

Some participants also hoped that the linkage of data across the healthcare system, and easier flows of live patient data between services, would improve their immediate care. They spoke about how it was difficult to know if their HIV status would be known about or taken into account in emergency treatment, and how it could be difficult to speak openly about their status in a new care setting. In addition, participants hoped that if HIV could be studied in conjunction with more illnesses, it would be more likely that "the clinician will know how to handle your issues" (FG2 BOR1) when a patient has HIV concurrent with one or more other conditions.

> "You might have an issue and you're rushed to the hospital and you're . . . you happen to be unconscious, it's easy to just . . . maybe, as I said, that they can know any previous illness or problems you had, all of them. So, that is in an advantage."
>
> (FG1 BOR1)

Participants were surprised when told about the lack of data flows by clinicians, and this caused them worry about the quality of their care:

> "She [Doctor] was saying that a lot of the patients that she sees are automatically assuming already that their data from the GUM clinic is automatically shared with the GP, and they're surprised that it isn't."
>
> (FG1 BOR1)

Because of the lack of joined up data, participants were aware of mistakes being made in their care because clinicians did not have the right data.

"All of these questions are now starting to pop into my head. . . about . . . how much kind of information is shared between all the various people that I see, the people in the GUM clinic, my GP, if I go into hospital for anything, and I'm beginning to find a kind of . . . the places where . . . I fall through. And that, and that worries me. Yeah, that worries me."

(FG1 BOR1)

"What can often happen is, just things like antidepressants, the GP could prescribe an anti-depressant that might contraindicate with their HIV medication. Unless . . . the patient is really savvy about knowing that that HIV antiretroviral combination has interactions then they could end up in a really, really sticky situation. And so, I think there are some real benefits around sharing data."

(FG2 BOR2)

Participants were therefore keen on HIV status data flowing in live patient information, as well as in extracted research databases, to improve the safety of their care. They felt exposed and at risk with the lack of joined up data and apparent ignorance of clinicians making decisions without knowing how their treatment for HIV might interact with new medications.

## Theme 2: Fear of discrimination

All participants had concerns about the protection of their data and its potential uses, these largely stemmed from low levels of trust that the healthcare system, and allied research systems, were working in their best interests and would host and use their data respectfully and securely.

"I think everybody agrees that the data research would be useful, but what we're getting is a general fear of how is that data protected, yeah?"

(FG1 BOR1)

**2.1 I've been discriminated against before, so I've lost trust in the health system.** Participants reported previous negative experiences resulting from the disclosure of their HIV status, and had experienced clinicians and organisations behaving in untrustworthy ways:

"There was a breach of my confidentiality around my HIV status and then a complete cover up and closing down when it came to trying to get information. Something went wrong but then also to try and address what goes wrong is not easily done."

(FG1 BOR2)

Because of previous breaches of trust, participants told us that they would be slow to trust new initiatives regarding their data, as they needed to build up trust over time. Trust would need to be *earned* by organisations, by continually demonstrating trustworthy behaviours, such as transparently keeping to safeguards which had been promised. Conversely, trust could be lost quickly by a single breach of the rules. This group of patients had had their trust in the system slowly chipped away over time, so would also need time and engagement to learn to trust the same system to look after their data.

"[It's not] something you can feel overnight, it's not something you can just wake up and let go, it is something that will affect a whole lot of time; you have to show this person that you are able to trust the organisation again."

(FG1 BOR2)

However, they acknowledged that to fully support data sharing, being able to trust in data safeguards described was very important for them to feel confident in allowing their data to be shared.

"I will start by saying, you know, trust is . . . the very important thing"

(FG2 BOR1)

While discussing the issue of trust and how this might be built over time with different organisations, participants demanded transparency over how third parties were given access to the data, and which organisations would have data. This was seen as especially important, given the increase to risk of re-identification with the possibility that additional sources of data, held by the organisation, could be matched into the de-identified data:

"There was a specific concern raised about if we do share this data, essentially, where does the data sharing stop and they wanted very clear statements on where this data could and couldn't be used, essentially, and because there was a concern that if a third party was able to access linked data, then they have more information, that third party might be able to source other information from other places."

(FG2 main room moderator summary).

**2.2 I could be reidentified through leaks, breaches or hacking.** Participants were concerned that despite de-identification of data, and data being stored and analysed in secure data environments, there might still be a breach of security in which their data would be leaked and made public. Even with the information presentations describing proposed data security, participants did not feel reassured by the system for keeping data safe, as they knew of hacking incidents on apparently secure systems in the past. They were concerned that it would only take one breach for their identity as HIV positive to become known, which would have detrimental consequences. The problem of being re-identified was serious, because participants knew that once the information got out, it could never be taken back. The patient would then be faced with trying to deny it had anything to do with them, which would cause stress.

"But on the side of the disadvantage, it makes you kind of feel . . . like . . . there's, maybe there's going to be a breach somewhere and a little mistake, and all your information that's in it, is out there."

(FG1 BOR1)

"Well, I'm quite . . . okay with the benefits of sharing my data, but what is really disturbing me is . . . if I'm reidentified, if somehow the data has been hacked and it's linked back to me. Like is there any way I can . . . deny this or something?"

(FG1 BOR1)

"You wake up one morning, your phone has been hacked, you wake up one morning and your system has been breached, so you have to be careful of who and what kind of organisation is handling this data."

(FG1 BOR2)

Support for data sharing for research was therefore predicated on the data being "100% confidential" (FG2 BOR1).

"As far as the 100% confidential is given to me, that it will not be exposed and nothing will happen to it, I have no problem about that."

(FG2 BOR1)

Participants articulated their concerns about the consequences of their data being made public or being re-identified following a breach. Many participants described that they kept their HIV status a secret from the people around them: "*anyone who's living with HIV tries to lead a very confidential life*" (FG2 BOR1); and they would fear stigmatisation and discrimination from other people if their status was known.

"For me, my fear generally is about letting people [who] I don't want to have an idea about this thing, to know that it's linked to me, you know, it's my personal information and I wouldn't like it to be shared. So, like I said, discrimination is also one of it. I think that's the main reason we really do not want our data to be shared."

(FG1 BOR2)

Because of the stigma attached to HIV, participants were worried that if others knew their HIV status, they would make assumptions about them, which were underpinned by stereotypes but not actually true. This might lead to them being seen as "problem cases", characterised by their condition, not as individuals with rich and varied lives.

"But it can be really, really disempowering if I . . . I'm just worried that what could happen is that . . . as a group of people we're not seen as people, we're actually seen as . . . problem cases or, you know, that patient that's living with HIV, so they will be . . . they're bound to be x, y, and z because they're living with HIV."

(FG2 BOR2)

Participants were also worried that other characteristics, such as their ethnicity, would intersect to create additional healthcare inequality if their HIV status was widely known. This was because they had experienced discrimination based on other characteristics, and feared this would be even worse if it was discovered they had a stigmatising condition.

"There was also a discussion about how . . . you can be stigmatized and health care, not just for HIV, but also for other characteristics like your ethnicity and those things can kind of compound and join together."

(moderator summary, FG2 main room).

**2.3 Data might be used to support an unhelpful agenda.** Participants were concerned that if data on their HIV status was available, it might be used to "*push problems towards HIV*"

rather than use the data to better inform HIV care. Participants' previous experience of health-care made them worried that any additional symptoms or problems might be written off as HIV-related rather than properly investigated, or patients may be expected to put up with lesser treatment for other conditions because of their HIV status.

> "There's a cynical bit of me that kind of feels, so you're looking at it from the perspective of how HIV information could be better informed by linking it into other illnesses. From another point of view people may just use that, if it's just one, this person is positive or not positive, they may just use that information to cloud other issues. It's been very easy over a long period of years to push problems towards HIV. You could do that with numbers, it's not impossible.
>
> (FG1 BOR2)

Because of the lack of trust in both researcher and clinician motives, participants were concerned about who might be funding research projects and what the agenda of the funder might be. If a researcher was taking money from a pharmaceutical company they may be motivated to find results in favour of prescribing new drugs for example, which may be biased. In response to this risk, the participants wanted researchers to be transparent about their motivations and underlying funding.

> "Some study at a university will say well who's funding? Generally there are drug companies' money or somebody else's money and I think that needs to be much more explicit."
>
> (FG1 BOR2)

**2.4 Certain organisations are less trustworthy.** Because of participants' prior experiences with different organisations discriminating against them or behaving in untrustworthy ways, they expressed reservations about some organisations handling their data. Because of the differing past experiences, diverse views about which organisations were trustworthy were expressed in the groups.

While some participants felt that all organisations were equal in terms of having access to the data, "*don't think there is any organisations I wouldn't like to share my data to as far as . . . as it was shared with them*" (FG1 BOR1), other participants felt strongly that only certain organisations should be able to access data: "*I certainly have concerns about data being used by organisations that I have issues with and also just what the purpose of that data was*" (FG1 BOR2). Trustworthiness of the data user organisation was key, "*it should be a group of people that are trustworthy*" (FG3 BOR1), but again, which organisations were seen as trustworthy was likely to depend on prior experiences, meaning there was not a consensus.

Largely, the healthcare sector was seen as trustworthy, and their motivations for using the data were seen as likely to be centred in patient benefit, with the added component of the sector having "something to lose" if safeguards and rules were not followed.

> "My personal concerns these days about my diagnosis are very much about how, not the hospitals and medical practitioners handle stuff, but how it's handled by other organisations."
>
> (FG1 BOR2)

> "NHS and pharma companies are trusted organisations to manage data. Mainly because they have something to lose if things go wrong."
>
> (FG3 BOR2 moderator summary)

However, some participants had concerns that wider public sector organisations such as "social services" or local authorities were less likely to be trustworthy, due to lesser perceived security with the data, and patients being less familiar with the services:

"All of these systems are only as good as the time and people who are using them allow. Nothing is going to be entirely secure. . . . I can see when it's contained within an NHS trust or something but I start to get more concern when there are links with maybe local social services, that type of thing because I think the work practices and the type of work is very, very different."

(FG1 BOR2)

"I'm more familiar, say, with who I think will see my data within the hospital. I've got quite a lot of experience of different departments and the different people that make that up. In social services and the council it's not something that you have. . . You're not going in there as such, you don't meet as many people, so it's like these are unknown people to you."

(FG1 BOR2).

Public sector university and "research organisations" were trusted in one group:

"I think it should be narrowed down to just specifically research organisations and also narrow it down to statistical organisations."

(FG1 BOR2),

but less trusted in another:

"Less trust on Universities as they are perceived as less trustworthy and capable of managing data"

(FG3 BOR2 moderator summary)

The "government" as a recipient of the data was discussed at length in one breakout room, with some people feeling like data should not be shared with the government; "*I think it should remain in in a health care system, not the government.*" (FG3 BOR1) and others feeling like people were "*all citizens of the government, so the government has access to this data, to enable the government to know that we're here.*" (FG3 BOR1)

Underpinning this was the idea that the "government" or potentially civil service, was large, with many people potentially having access to the data, which reduced the security measures in place:

"Many people are working for the government, and I think the more people working for the government, they are the risks of the data. But if it is in a healthcare system where you know, maybe two or three people are responsible for the data, and you know it's very secure."

(FG3 BOR1)

Previously expressed issues of trust and a sense of living in secrecy point towards potential feelings of fear around having data shared with the government which might be seen as an overarching faceless authority over people, not having their best interests at heart.

Commercial organisations were discussed among several groups, along with whether participants would feel comfortable with these organisations accessing their data. Insurance companies came up several times, with participants describing how the legacy impact of HIV on length of life still came up when applying for insurance. This was perceived as an outdated or ignorant view and therefore discriminatory, meaning insurance companies could not be trusted. Particularly the motivations of any commercial company were distrusted, and, given their profit motive, their compliance with data safeguards was not trusted. Because of the experience of ongoing stigma from insurance or other financial companies, participants preferred their data not to be shared with these.

"We're getting statistics from you guys which say that we live exactly the same lives as everybody else and yet we're still having to go to insurance organisations that will provide us, as HIV positive people, with specific insurances, and . . . I guess that's not your problem, but I would be reluctant . . . you know . . . you know, any kind of organisations like that."

(FG1 BOR1)

"As a positive HIV person, you know, people see you like you'll die quick, you know, people see you like you don't have this long lifespan because of your [HIV] status. So, I think it's really important that some . . . organisations, like the finance, the finance aspect of it, shouldn't have access to the . . . you know, to our data."

(FG2 BOR1)

Possibly because of a lack of perceived transparency in commercial entities, the other concern was that commercial companies may hold their own financial databases, and therefore might match in the health data, increasing the risk of re-identification.

"Even in an . . . in a pseudo-anonymised form, a deidentified form, then be sold on to a health insurance company or a commercial entity that then could potentially link that with other data that they might have, maybe financial data, and be able to kind of do a match . . . matching process, that's . . . that's the scary thing I think."

(FG2 BOR2)

In contrast, pharmaceutical companies, while still commercial entities, were seen very differently from insurance companies. This was because participants acknowledged that progress in the health of PLWH was largely due to pharmaceutical developments. There was a reciprocity in the sense of data sharing with pharmaceuticals, that while the company may profit from access to the data, the profit for the patient was also clear; better treatments and better health for patients. This sense of reciprocity was absent from the discussions about insurance or financial companies.

"Pharmaceutical organisations. . . you know, they're the people that are actually keeping me alive so, maybe they should have some of this data. You know, maybe they do need to know what are the medications I'm on."

(FG1 BOR1)

"Pharmaceutical companies should have access to the data because they are going to be able to then target that drug on the groups who need it and work out who the groups are and which drugs, need to be created and what problems need to be solved."

(FG2 BOR1)

Lastly, sharing data with religious groups was disliked due to perceived stigma.

"I think religious groups shouldn't be given access to this information because talking about religious groups, your culture, and you can't really say things, like it can lead to stigmatisation, oh these people are not people of God, these are not people that can exist."

(FG1 BOR2)

### Theme 3: Accountability and awareness

Following discussions around the perceived risks, participants offered a number of suggestions for safeguards, training and culture change they would want to see if their HIV status was included in health datasets. Suggestions ranged through IT security to training around HIV for data users and involvement of PLWH in research.

**3.1 Reassure me that my data is secure, I can't be identified.**   Any data-sharing was only supported on the proviso that identifiers were removed from the data and it was "100% confidential". Identifiers and personal information should be stripped out, and data encrypted in various ways. This was a consensus among the groups, that PLWH should be reassured, and be able to trust, that their identities would never be known from the data.

"It has to be very secure and personal information shouldn't be shared."

(FG2 BOR1)

"Where you have people have access to this data, I was thinking an encryption should be added to the data."

(FG1 BOR2)

Where personal information would be shared, one participant accurately demanded this should be with individual consent:

"So before personal information to be shared, I think the individual . . . needs to . . . consent . . . and should be informed, notified, before being shared out."

(FG3 BOR1)

However, what participants meant by "personal information" was not clear. While de-identified health data is, in law, not seen as "belonging" to the patient anymore, one participant felt the data was still personal even when identifiers had been removed, and this meant they should have a stake in what uses it was then put to.

"I strongly feel that whether or not your data is linked to your name or not it is still your data, it's information about you."

(FG1 BOR2)

It was not clear, therefore, that there was a consensus among the group that they supported securely de-identified data being shared without consent being given.

Additional computing-based securities were also important; participants felt that if IT systems were made very secure, this would give them reassurance about their data being shared.

"You know, we can send people to the moon. They get the IT in place, and then people will . . . be more willing to come on board with it, I think."

(FG1 BOR1)

**3.2 Reassure me that data users will follow the rules.** Participants identified that they might feel more secure about their data being shared, and data uses being appropriate, if the number of people who had access to the data were limited:

"The access of the data should be limited to certain persons or certain positions, and you get to control the more unforeseen circumstances or security breach."

(FG1 BOR2)

Limits could be applied by making the name of data users transparent, as well as their purposes; and by ensuring good data security training for all users.

"It would be good to have named people and very specific and obvious who is going to be accessing this data."

(FG1 BOR2)

Training would span both data security and governance, and the legacy of stigma around HIV (see subtheme 3.4).

"It will be somebody who is well trained to handle such confidential [data], on the monitors; somebody that has been trained on data handling, with data security on how to handle personal or confidential data."

(FG3 BOR1)

Some participants also suggested regular assessments of the data users' suitability for the role.

"You'd like to see regular reassessment of that to make sure that there's still somebody who is suitable to have that role."

(FG1 BOR2)

Trustworthiness could be mediated by both having transparency around rules and policies, but also clear enforcement of those rules. Participants were clear that there needed to be an accountability system so that data users who broke the rules would be punished.

"I think you do need to pin it down to named individuals, so that named people basically get the boot if they. . . I think it's the type of thing where you have to really be extremely punitive if mistakes are made because I think otherwise you just don't get people's trust and you're asking a lot of people to trust in all of these quite difficult systems to understand."

(FG1 BOR2)

For participants, the breakdown in trust which would ensue if a data user broke the rules was so serious, and the potential consequences so extensive, that they felt that, the data user's

ability to access data should be taken away permanently, and possibly, the data user should be dismissed from their job:

> "I think the measures should be take away your licence from assessing data or suspending them from ever. . . because you make a mistake once, it's somebody's life, so they don't do the mistake again."

(FG1 BOR2)

> "I know it sounds a bit extreme but I think it should be a sackable offence. . . the potential is there for very serious problems for people and I just think once it has been breached you can't unbreach it."

(FG1 BOR2)

**3.3 Combat the legacy of misunderstanding about HIV.** Because of the legacy of stigma around HIV, and discrimination against PLWH, participants expected that users of the data should be "educated" about HIV and undergo some awareness training. The expectation would be that this level of training would improve data users' current level of understanding of HIV and therefore reduce discriminatory perceptions of the virus. This was linked to the fear that data users may have an agenda that was ultimately going to cause harm to PLWH (sub-theme 2.3) when analysing the data.

> "Perhaps a reasonable expectation of those using HIV data is that it is mandatory (if possible) that they undergo some form of awareness training. There are reasons why HIV data has been controversial (stigma and prejudice)".

(Email communication from participant)

> "What I'm trying to say is that an education piece needs to go alongside the sharing of data if you see what I mean, especially for primary care type thing, where I know from my own experience that there is a really poor understanding of HIV as a long-term manageable condition now."

(FG2 main room)

Training to ensure up-to-date awareness of HIV was seen as likely to be especially valuable for people in the healthcare, or possibly research, systems who did not care for people with HIV as part of their medical specialty:

> "There needs to be a massive education programme for . . . people working in primary care, GPs, nurses working in primary care, alongside wider education around HIV for people not working in the specialty of HIV in the NHS."

(FG2 BOR2)

It was felt that this would be necessary due to the progress made on keeping PLWH healthy for many years, which other parts of the health system may not have kept up with:

> "HIV is understood where it is today in 2022 as opposed to. . . something that was extremely life limiting for most people. So, I think it isn't necessarily a data safeguard, its more around general education for people working in the NHS around HIV."

(FG2 BOR2)

These comments can clearly be understood in the context (previously described) of participants having recurrent experiences of discrimination, ignorance, or their trust in healthcare services being broken. As well as an awareness-raising programme for data users, one breakout group suggested *"Data protectors should be diverse to ensure that minority groups are not disadvantaged."* (FG3 BOR2 moderator summary).

**3.4 Communicate transparently, and close the loop—We're stakeholders.**   Participants identified that PLWH, as data-subjects, would want to know *"how your data is protected"* (FG1 BOR1), with *"really transparent information available about how it was being kept safe"* (FG1 BOR2). This transparency would mean *"that people can feel reassured and trust the system that their data was properly protected"* (FG1 BOR2).

In terms of how to approach this public communication, participants suggested social media, radio and TV.

> "Literally talking about everything, you know, the study from the Facebook, through there, like I listen to radio, I've been listening to . . . you know, watch the TV, so you can . . . just go on YouTube, you can just go on Twitter and get the information you want."

> (FG2 BOR1)

In addition, participants suggested that communications should cover who has access to the data, how the data is going to be stored and for how long, what projects it is used for and how the public can access the results of the projects,

> "It will give them a set of information that oh, that there is some certain set of people, or some certain set of researchers that are interested in their wellbeing."

> (FG2 BOR1)

> "You want to know where the data is going to be kept you want to know, probably for how long that they going to store, is it going to be stored forever?"

> (FG3 main room moderator summary)

> "Clear information regarding . . . how data will be managed. How data will be used and how results of the use of data will be delivered"

> (FG3 BOR2 moderator summary)

These discussions were not limited to communications towards PLWH, but seemed to cover informing the general public about wider uses of data.

As a further suggestion to increase transparency around data uses and therefore to slowly build up trust, participants in the groups were keen to see their own medical data. Participants felt frustrated that they still could not see health data about themselves, while others would be able to use it for analysis.

> "If professionals can see that data from the hospitals and the GUM clinics then I can see it, but at the moment I'm being told that I can't see it, yeah? It's not possible for me to see it. So, it seems like there's a bit of a chicken and egg situation here"

> (FG1 BOR1)

Participants felt that if PLWH could have access to all their NHS data, they might be more "on board" with the idea of data sharing for research, because they would have more of a sense of what information that data contained, and how identifying it might be.

> "The more pertinent question should be about how we can get access to the data that we can't see, that is going to be shared within this format, you know. GUM clinic, from the hospital, is it going to be supplying data to this? But at the moment we can't access that. We can see our GP data, but we can't see our hospital data or our GUM data, and I think people are going to understand and be more on board with the process if they're allowed to see the records that are being held for them."

> (FG1 BOR2)

Lastly participants wanted to know when their data was being used and for what purposes, and to be notified of results of studies conducted using their data. This level of transparency in the research process would build confidence that research was happening for public good, and thus, hopefully, over time build up trust. It would also mean that patients would feel they had a stake in the process of their data being used for research, as they could see what benefits people like them might experience, given new insights generated by the research.

> "I [would] feel more confident if I've been notified, then I can actually see what kind of information they're looking at, at that time. So, maybe whenever an institution or an organisation, anyone wants to access my data, I should get a notification on my phone."

> (FG2 BOR1)

> "I would love to see the results on whatever the outcome, or whatever the data has been used for . . . I would love to see the outcome of data that are being used."

> (FG3 main room)

**3.5 No research about us without us.**   Congruent with the above theme which posited PLWH as core stakeholders in the research, participants indicated that they would be more comfortable with sharing their data if there was "*significant and meaningful patient involvement in that kind of a thing.*" (FG1 BOR2). Particularly, they expected lived-experience involvement on panels or committees tasked with approving data usage proposals.

> "There should be a lay person who is living with HIV on that panel essentially providing insight, with that application, as to what are the pros and benefits to that person accessing that data."

> (FG1 BOR2)

People with lived experience would be able to scrutinise the proposals, ask questions about researchers' motivations, and suggest topics which would be more important to people living with HIV.

> "Those people who actually living with the condition [should] be able to, you know, very quickly, say, well, why are you asking this research question? Why are you doing it like that? Actually, these are the most important things that we're facing, actually can you include that as a research question?"

(FG2 main discussion)

This kind of involvement was seen to offer some kind of protection against research which might be well-meaning but would result in unintended adverse consequences for PLWH as a group, perhaps because it was poorly informed or did not understand the experience of living with HIV.

## Discussion

Our study aimed to explore the views of PLWH on sharing HIV status data in routinely collected health and care databases in England, and if they agreed with data sharing in theory, to explore what safeguards they would like to see implemented. We found an overarching theme centred around trust–with participants expressing how their trust in health and wider systems had been lost due to previous discriminatory experiences, and that while study participants were keen to see more research around HIV, there needed to be acknowledgement of lack of trust and a commitment to trust-building activities and trustworthy behaviour to regain support for data-sharing. Participants could see the potential benefits of HIV status being shared with wider health data for the purposes of better research and improved care, but support for data-sharing was predicated on it being 100% confidential, with no possible chance of re-identification. Participants told us that there would need to be full transparency about data flows and uses, and a slow building of trust that data safeguards would be fully enforced. Participants were keen to see research on mental health, quality of life, ageing and joining up care for long-term conditions in patients with HIV. They wanted to be seen as full stakeholders in the research process, with involvement in setting the research questions, ability to see their own data, and receiving information about the results of research studies.

Concerns about HIV status data being shared stemmed from prior poor experiences, and were largely centred concerns about data leaks or breaches which could lead to re-identification. There was also concern about stigma and discrimination for the HIV community which could result from research carried out by people who poorly understood or were not up to date on the condition. Participants identified a need for longer-term educational activities to address a lack of understanding and continued stigma around HIV which was perpetuated within healthcare and research systems. All of these issues are consistent concerns appearing in the findings of other HIV research and literature. While many themes discussed in this study also map onto previous studies investigating data sharing with the general population [21, 22], the need to build trust over time appeared more important to PLWH than other surveyed population groups. This was likely because of this group's experience of discrimination in many parts of society.

While this is the first study to ask PLWH about sharing HIV status data in databases of NHS patient records, previous literature has explored views of PLWH on research participation. These studies show similar concerns and safeguard preferences as our study. In previous studies, PLWH support more research happening and want more research on issues affecting their everyday lives. This has included wanting more research on differences between normal ageing and ageing with HIV, mental health issues, neurological symptoms, quality of life, HIV-related comorbidities, efficacy of alternative and complementary medicine, lifestyle behaviours, and access to end-of-life services [28]. Older people living with HIV were frustrated by the number of HIV-related research studies that excluded people with comorbidities, because many people living with HIV experience comorbidities such as depression, chronic illnesses, and neurological diseases [28].

In previous studies on research participation, HIV community members in the US had concerns about power imbalance with research participation, entrenched due to a lack of communication, these were made worse because PLWH felt like inferior or token members of research teams, and researchers demonstrated a lack of awareness about the needs of HIV study participants [29]. Other study participants felt that a risk of breach of confidentiality was a barrier to participation, especially if they belonged to a marginalised group [30]. These findings are similar to our study where participants wanted reassurances about data security, as well as involvement of PLWH on research teams and project screening committees, to ensure current stigmas and preconceptions about HIV did not become entrenched.

The safeguards proposed in our study such as ensuring data security, and including PLWH on research teams, map onto previous research with PLWH about research participation barriers. When interviewed about how to improve research participation and make taking part feel comfortable and safe for PLWH, older PLWH have previously stressed the importance of data confidentiality, and protecting participants' health information [28]. Very similarly to our study, elder HIV participants have suggested safeguards for research data, including masking identifiers through the use of identification numbers, using adequate technology to prevent cyber hacking, and informing participants how their data would be used and stored [28]. Participants in previous studies shared concerns about study biases arising from sponsored research, and suggested that conflicts of interest should be independently assessed by "representative" boards made of community members [28]. Additional safeguards in research have been suggested by black sexual minority men (BSMM), such as authentic engagement with the HIV community, and increased transparency of research processes, in order to build trust [31]. Our findings, while novel, are therefore congruent with previous research on PLWH's research participation and may well hold up in wider research on the topic.

## Strengths and limitations

This was, to the authors' knowledge, the first study to ask PLWH how they feel about their routine health data, including HIV status, being shared for research, and to invite them to suggest safeguards which would make them feel more comfortable with this data-sharing. One of the strengths of our study was the diverse sample, with many people from Black African, Black Caribbean and Mixed ethnicity, as well as participants reporting a range of sexual orientations. The format of giving information and inviting discussion was also a strength as it meant our participants could reason from a more informed viewpoint, having already heard about the different safeguards which are usually put in place around health data.

However, our study is limited by the relatively small sample size, the recruitment via HIV charities, the anonymity we allowed for participants during the Zoom focus groups, and the self-report nature of our demographic data collection. We cannot be sure all study participants were actually living with HIV, as this was self-reported. We noted a much increased recruitment rate after we drew more attention to the monetary voucher we were offering for participation, which may have drawn people to take part and state that they were eligible, when they actually were not. This issue is likely to affect online research most, such as web-based surveys and online discussions using video-conferencing. We did not screen participants (apart from by self-report) to check they lived in England or had a diagnosis of HIV. Because of this, for example, while aiming to recruit only in England, one of our participants was resident in Scotland which has a slightly different health data sharing set up.

After consideration, we decided it was worth taking a risk of inadvertently recruiting people who were not eligible, to increase participation of groups who experience the most marginalisation in society. Participants who take part in research are more likely to be health literate,

data literate, and engaged with their identities, in ways that other parts of the HIV community may not be, and thus our sample may represent a minority set of views. On the other hand, our sample is fairly unusual for a UK HIV study in terms of high numbers of participants with non-white ethnicity, English as a second language, and heterosexual orientation, suggesting it was not dominated by people who may "usually" take part in HIV research. More surveying of the UK HIV population on this issue would certainly be warranted from our results, acknowledging that the voices of some people living with HIV are seldom heard, and striving to achieve a representative sample. It is encouraging, despite these limitations, that our results are so consistent with previous research around barriers to research participation for PLWH, and we feel hopeful that further research would confirm and enrich our findings.

## Future directions

Changing rules and practices around data sharing in the English NHS is not a quick or straightforward process. However, as far as we know, there is no legal impediment to sharing HIV diagnosis codes to be used for secondary purposes, within de-identified healthcare data in the UK; the routine removal of these appears to be a legacy practice from when such information was considered special category data.

It must be noted, however, that several NHS data sharing initiatives have previously failed because of a lack of communication around them and subsequent patient and public distrust of the initiative [32]. An example of this was the 2013–14 Care. data project, which was postponed and then shelved after a public outcry about the safety of de-identified data, the contracting of a private company to manage the data extraction, and general misgivings about the government's motivations and intention to sell public data [33]. This legacy of failure of transparency and trustworthiness with NHS data projects hangs over all new initiatives [34]. Our findings suggest that in order to start sharing HIV data for research in a supported and sustainable way, there needs to be a commitment to clear trust-building activities such as open communications, data transparency, state of the art technology for security, education about HIV in both health and research systems, and thoroughly embedded and valued patient involvement. Without these in place, PLWH's trust in the system and, therefore, the whole endeavour, will continue to be low.

## Conclusions

Having engaged deliberatively with PLWH from a range of backgrounds, we suggest that, if well planned and transparent, an initiative to work with the HIV community to discuss sharing HIV status data in NHS research databases could start. This should be seen as a long-term relationship-building engagement process, to foster trust. Some trustworthy values that could underpin the initiative would include high-tech data security, limited access and clear accountability for data users, meaningful involvement of people living with HIV, alongside education about HIV to reduce stigma. As core stakeholders, PLWH want to know what is happening with their data and what research and discoveries have been made using it. Once these assurances have been put in place, it is likely that PLWH would support their HIV status data being shared in the hope of better research being undertaken to improve services and ultimately their health and wellbeing.

## Supporting information

**S1 File. Sample screenshots of focus group presentation slides.**
(PDF)

**S2 File. Human subjects research checklist.**
(DOCX)

## Acknowledgments

We would like to thank AIDSmap, Terrence Higgins Trust and the National Aids Trust for facilitating the recruitment of participants by allowing us to distribute flyers with information through their newsletters.

## Author Contributions

**Conceptualization:** Elizabeth Ford, Michael Smith, Jaime Vera.

**Data curation:** Katie Goddard, Michael Smith.

**Formal analysis:** Elizabeth Ford, Katie Goddard.

**Funding acquisition:** Elizabeth Ford, Jaime Vera.

**Investigation:** Elizabeth Ford, Michael Smith.

**Methodology:** Elizabeth Ford, Jaime Vera.

**Project administration:** Michael Smith.

**Supervision:** Elizabeth Ford.

**Writing – original draft:** Elizabeth Ford, Katie Goddard.

**Writing – review & editing:** Elizabeth Ford, Katie Goddard, Michael Smith, Jaime Vera.

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
