## [Decision Letter · Decision Letter 0]

6 May 2024

PONE-D-24-06042“I don’t see a reason why we should be hidden from view”: Views of people living with HIV on sharing HIV status data in routinely collected health and care databases in EnglandPLOS ONE

Dear Dr. Ford,

Thank you for submitting your manuscript to PLOS ONE. This is an interesting article that makes a novel contribution to the field. After careful consideration, we feel that it has merit but does not fully meet PLOS ONE’s publication criteria as it currently stands. Therefore, we invite you to submit a revised version of the manuscript that addresses the points raised during the review process.

We look forward to receiving your revised manuscript.

Kind regards,

Craig Donnachie, Ph.D

Academic Editor

PLOS ONE

Journal Requirements:

2. Please expand the acronym “NIHR” (as indicated in your financial disclosure) so that it states the name of your funders in full.

**Additional Editor Comments:**

We invite you to revise your manuscript by responding to reviewer comments below. Please take note with regards to R2 points, especially with regards to the sampling strategy employed and the potential wider ramifications. Greater clarity if needed with regards to the analytical approach employed. Further attention is needed with regards to the presentation of the analysis specifically the narrative pertaining to the data (see R1 and R2 points).

Reviewers' comments:

Reviewer's Responses to Questions

**Comments to the Author**

1. Is the manuscript technically sound, and do the data support the conclusions?

Reviewer #1: Yes

Reviewer #2: Partly

2. Has the statistical analysis been performed appropriately and rigorously? 

Reviewer #1: N/A

Reviewer #2: N/A

3. Have the authors made all data underlying the findings in their manuscript fully available?

Reviewer #1: Yes

Reviewer #2: No

4. Is the manuscript presented in an intelligible fashion and written in standard English?

Reviewer #1: Yes

Reviewer #2: Yes

5. Review Comments to the Author

Reviewer #1: The authors have presented a novel and interesting article, which at the moment is in a very topical area. The study presents original and novel research, with the rationale for the study clearly presented. The study gives a well-rounded introduction which introduces the topic and gap in the knowledge well. The approach to the focus groups is interesting and seems to have worked well in encouraging discussion amongst participants. More information could have been given on the presentation contents (ie were images included or text only?), however this is a minor issue and doesn’t impact the overall article. The findings are consistent with other studies in this area, particularly in relation to the fear of stigma, reluctance to share data and the mistrust of the UK health system. The thematic analysis has been systematically conducted with detailed themes, backed up thoroughly by the data. The ethical considerations have been outlined at every point and the ethical consideration has been carried through to the availability of data for the journal also. The authors have clearly outlined the reasons for limited access to the data, based on meeting the criteria for accessing sensitive data. This is commendable, particularly in light of the topic and findings of the study. Overall, the authors have presented a well-written and interesting article which deserves publication!

Reviewer #2: Major: The paper is interesting but it should be much clearer that this is convenience sample that is recruited via HIV charities and not representative of the vast majority of PLHIV. Any conclusions should be measured and relate to the specificity of this sample. The partipcants are likely to be health literate, data literate and engaged with their identities in ways others will not be. These issues should be clear from the title, abstract and limitations sections.

Major: more detail is needed about the type of thematic analysis used. Which version of Braun and Clerks approach was employed. Reflexive, latent, semantic, deductive etc. Can the authors clarify how what they present is thematic analysis and not content analysis? The themes look very closely related to the deliberate topics raised etc. Maybe in Table 3 you could visualise anything that was generated from the participants rather than your schedule?

Major: the presentation of analysis is not pitched well. There is far too much data and not enough narrative about the analysis (this makes it look under-analysed and quite like content analysis with an excess of quotes). I would recommend that the results are re-written with a strong authorial voice about the findings of the analysis illustrated by selected illustrative extracts than the large volume of largely descriptive and slightly repetitive data extracts presented now. I think the reader wants an account of analysis not the presentation of excess data. As a rule any duplication (across extracts) or between authors account and the data presented should be removed.

I feel the discussion should also indicate the need to further explore the issues within representative samples of PLHIV in the UK, and the need to explore the potential of mixed methods etc. It would be a great opportunity to discuss who does get to speak for PLHIV in the UK. I am also interested in the authors thoughts about the veracity of their sample. I am aware that for online research there has been an increase in organised engagment with UK research using VPNs to appear within the UK but participation can be fake. Were they convinced of the authenticity of their sample (sorry if this is a weird thing to ask)

Minor: is the study in UK, England or Scotland? Please fid a consistent approach.

6. PLOS authors have the option to publish the peer review history of their article (what does this mean?). If published, this will include your full peer review and any attached files.

Reviewer #1: No

Reviewer #2: **Yes: **Paul Flowers

---

## [Author Response · Author response to Decision Letter 0]

24 Sep 2024

Editor Please take note with regards to R2 points, especially with regards to 

- the sampling strategy employed and the potential wider ramifications. 

- Greater clarity if needed with regards to the analytical approach employed. 

- Further attention is needed with regards to the presentation of the analysis specifically the narrative pertaining to the data (see R1 and R2 points). 

>>Many thanks for your careful consideration of our paper. We have added further comments on the sample in the discussion section (p34-35). 

We have thoroughly re-written the presentation of the findings according to the reviewers’ points and feel the results section is much improved (p10-30). In the methods we have also explained further about the qualitative analysis approach (p7). 

Reviewer 1 The authors have presented a novel and interesting article, which at the moment is in a very topical area. The study presents original and novel research, with the rationale for the study clearly presented. The study gives a well-rounded introduction which introduces the topic and gap in the knowledge well. The approach to the focus groups is interesting and seems to have worked well in encouraging discussion amongst participants. 

>>Thank you – no changes made.

More information could have been given on the presentation contents (ie were images included or text only?), however this is a minor issue and doesn’t impact the overall article. 

>>We have supplied a sample of slide screen shots as supplementary materials. 

The findings are consistent with other studies in this area, particularly in relation to the fear of stigma, reluctance to share data and the mistrust of the UK health system. The thematic analysis has been systematically conducted with detailed themes, backed up thoroughly by the data. The ethical considerations have been outlined at every point and the ethical consideration has been carried through to the availability of data for the journal also. The authors have clearly outlined the reasons for limited access to the data, based on meeting the criteria for accessing sensitive data. This is commendable, particularly in light of the topic and findings of the study. Overall, the authors have presented a well-written and interesting article which deserves publication! 

>>Thank you – no changes made. 

Reviewer 2 The paper is interesting but it should be much clearer that this is convenience sample that is recruited via HIV charities and not representative of the vast majority of PLHIV. Any conclusions should be measured and relate to the specificity of this sample. 

>>The words “convenience sample” have been added in Title (page 1), Abstract (page2) and Methods (page 5). 

The partipocants are likely to be health literate, data literate and engaged with their identities in ways others will not be. These issues should be clear from the title, abstract and limitations sections. 

>>We have added this caveat to the discussion section (page 34-35)

More detail is needed about the type of thematic analysis used. Which version of Braun and Clerks approach was employed. Reflexive, latent, semantic, deductive etc. Can the authors clarify how what they present is thematic analysis and not content analysis? The themes look very closely related to the deliberate topics raised etc. Maybe in Table 3 you could visualise anything that was generated from the participants rather than your schedule? 

>>We thank the reviewer for their constructive review which helped us reconsider our analysis completely. We agree it was previously “undercooked”. We spent more time reflecting on the data, revisiting the codes and the themes and adding more interpretation and layers of meaning. We have added information about the further analytic process as well as some self-reflection on the analysis it produced in the methods section (p7). We have produced a figure which depicts the new theme structure rather than the previous table which looked like a list (p10). 

The presentation of analysis is not pitched well. There is far too much data and not enough narrative about the analysis (this makes it look under-analysed and quite like content analysis with an excess of quotes). I would recommend that the results are re-written with a strong authorial voice about the findings of the analysis illustrated by selected illustrative extracts than the large volume of largely descriptive and slightly repetitive data extracts presented now. I think the reader wants an account of analysis not the presentation of excess data. 

>>We have spent a good amount of time adding an extra layer of interpretative reflection and reformation of themes. We have looked for coherent and meaningful phenomena underpinning the views expressed in the groups and feel this new write up reflects the voice and experiences of the participant group in a deeper way. See the whole results section (p10-30)

As a rule any duplication (across extracts) or between authors account and the data presented should be removed. 

>>We have removed various quotes where the same point was being made multiple times. 

I feel the discussion should also indicate the need to further explore the issues within representative samples of PLHIV in the UK, and the need to explore the potential of mixed methods etc. It would be a great opportunity to discuss who does get to speak for PLHIV in the UK. 

>>We have added to the discussion section on limitations and suggested some further research approaches which could be used to extend or confirm our findings. (P34-35)

I am also interested in the authors thoughts about the veracity of their sample. I am aware that for online research there has been an increase in organised engagment with UK research using VPNs to appear within the UK but participation can be fake. Were they convinced of the authenticity of their sample (sorry if this is a weird thing to ask) 

>>We’ve added a few sentences on how online samples can be made up of participants who are not actually eligible (p34-35). 

Is the study in UK, England or Scotland? Please fid a consistent approach. 

>>We’ve highlighted the limitation that while we aimed for an English sample, one of our participants was resident in Scotland. Their data was not removed. (p34-35)

---

## [Decision Letter · Decision Letter 1]

17 Dec 2024

“I don’t see a reason why we should be hidden from view”: Views of a convenience sample of people living with HIV on sharing HIV status data in routinely collected health and care databases in England

PONE-D-24-06042R1

Dear Dr. Ford,

We’re pleased to inform you that your manuscript has been judged scientifically suitable for publication and will be formally accepted for publication once it meets all outstanding technical requirements.

Kind regards,

Sebastian Suarez Fuller, PhD

Academic Editor

PLOS ONE

Additional Editor Comments (optional):

Reviewers' comments:

Reviewer's Responses to Questions

**Comments to the Author**

1. If the authors have adequately addressed your comments raised in a previous round of review and you feel that this manuscript is now acceptable for publication, you may indicate that here to bypass the “Comments to the Author” section, enter your conflict of interest statement in the “Confidential to Editor” section, and submit your "Accept" recommendation.

Reviewer #1: All comments have been addressed

2. Is the manuscript technically sound, and do the data support the conclusions?

Reviewer #1: Yes

3. Has the statistical analysis been performed appropriately and rigorously? 

Reviewer #1: N/A

4. Have the authors made all data underlying the findings in their manuscript fully available?

Reviewer #1: Yes

5. Is the manuscript presented in an intelligible fashion and written in standard English?

Reviewer #1: Yes

6. Review Comments to the Author

Reviewer #1: (No Response)

7. PLOS authors have the option to publish the peer review history of their article (what does this mean?). If published, this will include your full peer review and any attached files.

Reviewer #1: No

---

## [Editor Report · Acceptance letter]

14 Jan 2025

PONE-D-24-06042R1 

PLOS ONE

Dear Dr. Ford, 

I'm pleased to inform you that your manuscript has been deemed suitable for publication in PLOS ONE. Congratulations! Your manuscript is now being handed over to our production team.

Kind regards, 

on behalf of

Dr. Sebastian Suarez Fuller 

Academic Editor

PLOS ONE